# Prognostic value of composite inflammatory markers in patients with chronic obstructive pulmonary disease: A retrospective cohort study based on the MIMIC-IV database

**Xingxing Liu**[1], **Yikun Guo**[2,3], **Wensheng Qi**[1]*

**1** Guanganmen Hospital Affiliated to China Academy of Chinese Medical Sciences, Xicheng District, Beijing, China, **2** Dongzhimen Hospital, Beijing University of Chinese Medicine, Dongcheng District, Beijing, China, **3** Beijing University of Chinese medicine, Chao Yang District, Beijing, China

* 8203803831@163.com

**Data Availability Statement:** The clinical data related to the MIMIC database obtained in this paper is only a small part of the MIMIC database. The relevant data can be publicly obtained on the

## Abstract

Chronic obstructive pulmonary disease (COPD) is a chronic inflammatory lung disease, and inflammation plays a key role in the pathogenesis of COPD. The aim of this study is to investigate the association between systemic immune inflammation index (SII), systemic inflammatory response index (SIRI),pan-immune inflammation value (PIV), neutrophil-to-lymphocyte ratio (NLR), and platelet-to-lymphocyte ratio (PLR) and all-cause mortality in patients with chronic obstructive pulmonary disease (COPD), and to evaluate the effect of composite inflammatory markers on the prognosis of COPD patients. We obtained data on COPD patients from the Medical Information Mart for Intensive Care (MIMIC) -IV database and divided patients into four groups based on quartiles of baseline levels of inflammatory markers, The primary outcomes were in-hospital and ICU mortality. We comprehensively explored the association between composite inflammatory markers and mortality in patients with COPD using restricted cubic splints (RCS), COX proportional hazards regression models, Kaplan-Meier curves, receiver operating characteristic (ROC), and subgroup analyses. A total of 1234 COPD patients were included in this study. RCS results showed that SII, SIRI, PLR, PIV and NLR were positively and non-linearly correlated with the increased risk of in-hospital mortality in COPD patients. Multivariate COX regression analysis showed that compound inflammatory markers were independent risk factors for in-hospital mortality in COPD patients. The KM curve results showed that COPD patients with higher SII, SIRI, PLR and PIV had a significantly lower survival probability. 5 kinds of compound between inflammatory markers and mortality in patients with COPD is related to nonlinear correlation, can increase the risk of mortality in patients with COPD is a risk factor for the prognosis of patients with COPD, and may serve as potential biomarkers for clinical COPD risk stratification and treatment management in critical patients.

following websites, and the original data contained in the paper is also provided in S3 File. All data in the paper are based on public data for secondary data analysis without ethical approval or primary data, and all data can be accessed at the following websites: MIMIC-IV database, (https://physionet.org/content/mimiciv/2.2/).

**Funding:** This research was financially supported by the Key Collaborative Research Project of Science and Technology Innovation Engineering of the Chinese Academy of Traditional Chinese Medicine (CI2022C005). We sincerely thank the sponsor for their support in deciding to publish this article. We also deeply appreciate their financial support and guidance throughout the preparation of this paper.

**Competing interests:** The authors have declared that no competing interests exist.

## Introduction

COPD is a common and persistent inflammatory disease of the airway, including persistent respiratory symptoms and irreversible airflow limitation [1]. Epidemiological studies have shown that, because of its high morbidity and mortality of COPD ranks third in the global disease cause of death, behind only ischemic heart disease and cerebrovascular disease, the global medical resources and social economy produced a heavy burden [2]. The pathogenesis of COPD complicated, involving the combination [3] of environmental and genetic factors. Therefore, search for new biomarkers to predict clinical outcomes can help clinicians to identify high-risk patients and early prevention strategy implementation, can reduce aderse outcome in patients with COPD, it is vital for the prevention, management and treatment of COPD.

The main characteristic of COPD is persistent airway inflammation and immune dysfunction [4–6]. Related studies have found that a variety of inflammation-related composite blood markers have potential clinical utility for COPD disease progression, prognosis and survival. SII, SIRI, PIV, NLR and PLR are five novel composite inflammatory markers, involving neutrophils, monocytes, platelets and lymphocytes. Closely associated with inflammation in the body, the immune regulation. Composite inflammatory markers are not only cheap and easy to obtain, but also more accurate than single biological markers to reflect the inflammatory response and immune regulation in human pathological conditions.However, there is no study to explore the clinical value of SII, SIRI, PIV, NLR and PLR in the prognosis of COPD patients with large samples.

In this study, the clinical data of critically ill COPD patients in MIMIC-IV database were obtained to analyze the potential relationship between five composite inflammatory markers, including SII, SIRI, PLR, PIV and NLR, and all-cause mortality in critically ill COPD patients, which may establish them as risk stratification tools for COPD patients and risk factors for predicting prognosis.

## Materials and methods

### Data sources

Of this study were retrospectively analyzed the data and Information are from Intensive Care medicine Information market (Medical Information Mart for Intensive Care, MIMIC—IV version 2.2). MIMIC-IV is a large, free-access database that includes data [7] on patients admitted to the intensive care unit (ICU) of Beth Israel Deaconess Medical Center (BIDMC) between 2008 and 2019. Patient information recorded in the MIMIC-IV database includes patient vital status, demographics, laboratory tests, medications, vital signs, and other detailed data. For this study, ethical approval was granted by the institutional review boards at MIT and BIDMC. This study is to MIMIC—IV secondary analysis of the database, so there is no need to provide ethical approval and related documents. In order to ensure the confidentiality of patient data, all personal information by using random code instead of status for anonymous, no patient consent and ethical approval. Author (Yikun Guo) (certificate number: 62099487) to complete the National Institutes of Health (National Institutes of Health) related training course, the database access permission.

### Study population

We screened COPD patients from the MIMIC-IV database according to ICD-9 (49320–49322) and ICD-10 codes (J441, J449) (Fig 1). Exclusions are marked as follows: (1) under 18 years of age;(2) ICU length of stay <24 hours; (23) for patients with multiple hospitalizations,

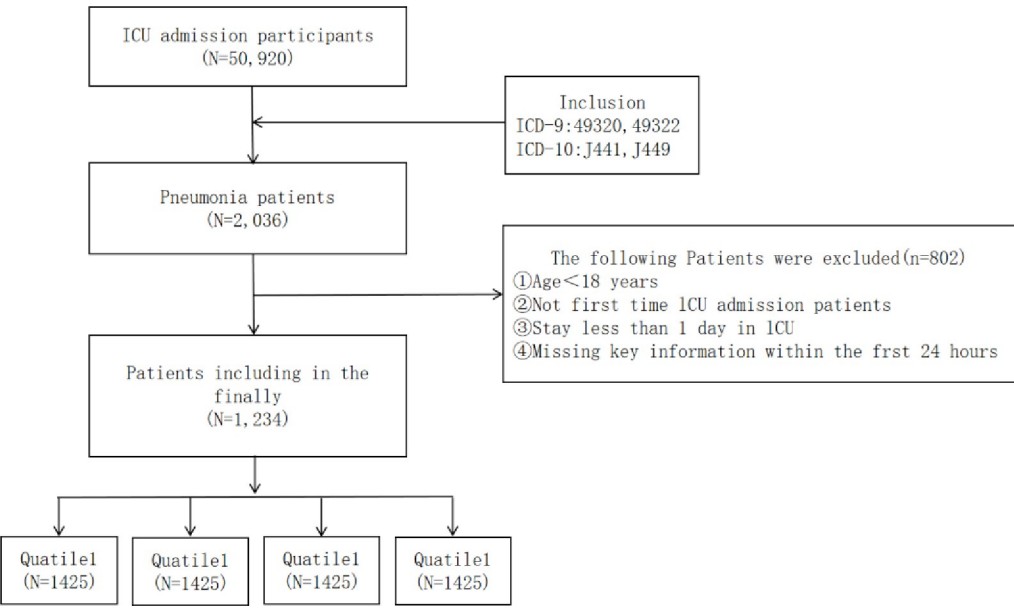

**Fig 1. Flow chart of patient selection.**

only the data of first admission were extracted; And (4) patients without key data (neutrophil, lymphocyte, monocyte, and platelet counts) within 24 hours after admission.

## Data extraction

In this study, PostgreSQL software (version 13.7.0) was used to obtain data from MIMIC-IV database using structured query language. The extracted variables were as follows: (1) Demographic information: age, gender, race, and weight. (2) Vital signs: heart rate (HR), mean blood pressure (MBP), systolic blood pressure (SBP), diastolic blood pressure (DBP), respiratory rate (RR), body temperature, blood oxygen saturation (SPO2). (3) Laboratory parameters: White blood cell count (WBC), red blood cell count (RBC), central granulocyte count, lymphocyte count, monocyte count, platelet count (PLT), hemoglobin (HGB), chloride (Cl), potassium (K), sodium (Na), calcium (Ca), glucose (Glu), anion gap (AG), creatinine (Cr), blood urea nitrogen (BUN), activated partial thromboplastin time (APTT). (4) Complications: hypertension, myocardial infarction, malignant tumor, pneumonia, atrial fibrillation (AF), renal failure and coronary heart disease (CHD). (5) Scoring system: sequential organ failure assessment (SOFA) score, simplified acute physiology score II (SAPSII), acute physiology and chronic health evaluation score II (APSIII), Oxford acute severity of illness score (OASIS), Glasgow coma Scale (GCS), Charlson composite index score (CCI).

## Group and outcome

The systemic immune inflammation index (SII) was calculated using the formula: (platelet count × neutrophil count)/lymphocyte count; Systemic inflammatory response index (SIRI): (neutrophils × monocytes)/lymphocytes; Pan-immune inflammation index (PIV): (neutrophil count × monocyte count × platelet count)/lymphocyte count; Neutrophil/lymphocyte ratio (NLR): neutrophil count/lymphocyte count; Platelet/lymphocyte ratio (PLR): platelet count/lymphocyte count.

Patients were divided into four groups according to the quartile of inflammatory markers. The primary outcomes were in-hospital and ICU mortality, and the secondary outcomes were length of hospital stay and ICU length of stay.

## Statistical analysis

We assessed the normal distribution of continuous variables using the Kolmogorov-Smirnov test. Normally distributed continuous variables were described as means ± standard deviations and were compared between the two groups with the use of student's t-test. Categorical variables were presented as numbers and percentages, and differences between groups were compared with the use of the chi-square test.

We analyzed the association between inflammatory markers and all-cause mortality in COPD patients using Kaplan-Meier (KM) curves and Cox proportional hazards regression models, expressed as hazard ratio (HR) and 95% confidence interval (CI). Four models were constructed based on the variables included in the COX regression analysis, without any adjustment in model I; Model II was adjusted for age, gender, weight, race, heart rate, respiratory rate, body temperature, MBP, SBP, DBP, SPO2, SOFA, SAPSII, APSIII, OASIS, GCS, and CCI. Model III was further adjusted for comorbidities, including hypertension, myocardial infarction, cancer, pneumonia, AF, renal failure, and CHD. Model on the basis of model III, IV for the WBC, RBC, center granulocytes, lymphocyte, monocyte, PLT, HGB, Cl, K, Na, Ca, Cr, BUN, Glu, AG, APTT laboratory indexes such as to adjust. In addition, restricted cubic spline plots (RCS) were used to evaluate the nonlinear relationship between inflammatory markers and all-cause mortality, and receiver operating characteristic (ROC) was used to compare the predictive ability of five inflammatory markers for the risk of death in COPD patients. Each statistical test was performed with the use of a two-tailed design. R, version 4.2.2, was used for statistical analyses, and P values of 0.05 or less were considered to indicate statistical significance.

## Results

### Baseline characteristics

A total of 1234 patients with COPD critically ill were screened from the MIMIC-IV database according to strict inclusion and exclusion criteria, including 677 (54.9%) men and 557 (45.1%) women, with a mean age of 71.3 years. The in-hospital and ICU mortality rates were 14.83% and 9.89%, respectively. The mean length of hospital stay and ICU stay were 14.5 days and 6.8 days, respectively.

The baseline characteristics of the in-hospital survival of the COPD patients are shown in Table 1 and included 1051 patients who survived and 183 patients who died. Among the patients in the death group, the proportion of women was higher, the patients were older, lighter, often complicated with acute renal failure and pneumonia, and the severity of illness score was higher, and the levels of Glu, APTT, Cr and BUN were higher. Compared with the group of patients survival, death group of patients with SII, SIRI, PIV, PLR and NLR were significantly increased ($P < 0.05$). S1 Table in S1 File shows the baseline survival characteristics of patients with COPD admitted to the ICU. Patients in the death group had a higher white blood cell level, and the rest of the baseline characteristics were similar to those in Table 1. The levels of SII, SIRI, PIV, PLR and NLR in the death group were also significantly higher than those in the survival group ($P < 0.05$).

### Relationship between inflammatory markers and COPD mortality

We assessed the association between inflammatory markers and all-cause mortality by RCS. RGC results (Fig 2), SII ($P = 0.001$), SIRI ($P < 0.001$), PIV ($P = 0.013$), PLR ($P = 0.001$) and

**Table 1. Baseline characteristics of the survival status of hospitalized patients with COPD.**

| Variable | | | Survival(N = 1051) | Death(N = 183) | Total(N = 1234) | P-value |
|---|---|---|---|---|---|---|
| **Demographic** | | | | | | |
| | Gender,N(%) | | | | | 0.038 |
| | | Female | 461 (43.9%) | 96 (52.5%) | 557 (45.1%) | |
| | | Male | 590 (56.1%) | 87 (47.5%) | 677 (54.9%) | |
| | Race,N(%) | | | | | < 0.001 |
| | | Black | 62 (5.9%) | 5 (2.7%) | 67 (5.4%) | |
| | | Other | 23 (2.2%) | 3 (1.6%) | 26 (2.1%) | |
| | | Unkonwn | 171 (16.3%) | 59 (32.2%) | 230 (18.6%) | |
| | | White | 780 (74.2%) | 115 (62.8%) | 895 (72.5%) | |
| | | Yellow | 15 (1.4%) | 1 (0.5%) | 16 (1.3%) | |
| | Age,(year) | | 70.8 (60.3, 81.3) | 74.0 (62.8, 85.2) | 71.3 (60.6, 82.0) | < 0.001 |
| | weight(Kg) | | 84.4 (59.6, 109.2) | 77.9 (54.0, 101.8) | 83.4 (58.7, 108.1) | 0.001 |
| **Vital signs** | | | | | | |
| | HR,(times/min) | | 88.7 (69.8, 107.6) | 93.5 (70.3, 116.7) | 89.4 (69.7, 99.1) | 0.009 |
| | RR,(times/min) | | 19.7 (13.0, 26.4) | 20.8 (14.2, 27.4) | 19.9 (13.2, 26.6) | 0.047 |
| | Temperature,(°C) | | 36.8 (35.6, 38.0) | 36.8 (36.3, 37.3) | 36.8 (35.7, 37.9) | 0.402 |
| | DBP,(mmHg) | | 68.3 (49.9, 86.7) | 67.7 (48.3, 87.1) | 68.2 (49.7, 86.7) | 0.673 |
| | SBP,(mmHg) | | 119.8 (94.9, 144.7) | 117.3 (92.7, 141.9) | 119.5 (94.6, 144.4) | 0.207 |
| | MBP,(mmHg) | | 87.5 (70.4, 104.6) | 80.6 (61.3, 99.9) | 86.5 (73.7, 99.3) | 0.246 |
| | SPO2,(%) | | 96.6 (92.7, 100.5) | 95.7 (91.1, 100.3) | 96.5 (92.5, 100.5) | 0.013 |
| **Scoring systems** | | | | | | |
| | SOFA | | 5.4 (2.0, 8.8) | 8.3 (4.1, 12.5) | 5.9 (2.2, 9.6) | < 0.001 |
| | APSIII | | 44.8 (26.8, 62.8) | 62.9 (37.8, 88.0) | 47.5 (27.2, 67.8) | < 0.001 |
| | SAPSII | | 38.9 (26.6, 51.2) | 49.1 (34.6, 63.6) | 40.4 (27.2, 53.6) | < 0.001 |
| | OASIS | | 32.7 (25.0, 40.4) | 38.1 (28.8, 47.4) | 33.5 (25.3, 41.7) | < 0.001 |
| | GCS | | 13.7 (11.1, 16.3) | 13.3 (10.3, 16.3) | 13.7 (11.1, 16.3) | 0.08 |
| | CCI | | 6.7 (4.0, 7.4) | 8.0 (5.2, 10.8) | 6.9 (4.1, 9.7) | < 0.001 |
| **Laboratory results** | | | | | | |
| | WBC,(K/UL) | | 14.0 (5.4, 22.8) | 13.6 (5.7, 21.5) | 13.9 (5.5, 25.3) | 0.653 |
| | RBC,(m/UL) | | 3.5 (2.7, 4.3) | 3.3 (2.5, 4.1) | 3.4 (2.6, 4.2) | 0.014 |
| | Neutrophils,(K/UL) | | 10.5 (3.2, 17.8) | 11.8 (4.1, 19.5) | 10.7 (3.3, 18.1) | 0.03 |
| | lymphocyte,(K/UL) | | 1.8 (0.5, 3.1) | 0.9 (0.1, 1.7) | 1.7 (0.5, 2.9) | 0.007 |
| | monocyte,(K/UL) | | 6.6 (2.0, 11.2) | 6.8 (1.0, 12.6) | 6.6 (1.9, 11.3) | 0.557 |
| | PLT,(K/UL) | | 191.2 (95.8, 286.6) | 187.1 (88.5, 285.7) | 190.6 (94.8, 286.4) | 0.589 |
| | HGB,(g/dL) | | 10.2 (8.0, 12.4) | 9.9 (7.5, 12.3) | 10.2 (8.0, 12.4) | 0.136 |
| | SII | | 4349.9 (7.8,1722697) | 4331.1 (577.1, 8085.1) | 4363.7 (1568.8, 7158.6) | 0.001 |
| | SIRI | | 76.2 (10.5, 141.9) | 137.2 (25.0, 137.2) | 89.4 (16.8, 162.0) | < 0.001 |
| | PIV | | 18744.2 (5336.8, 32151.6) | 23010.7 (2998.0, 43023.4) | 19567.1 (5180.2, 33954) | < 0.001 |
| | NLR | | 16.4 (3.8, 29.0) | 21.2 (0.5, 41.9) | 17.4 (3.6, 31.2) | < 0.001 |
| | PLR | | 280.5 (44.4, 516.6) | 388.9 (49.3, 728.5) | 301.1 (49.7, 552.5) | < 0.001 |
| | Na,(mmol/L) | | 138.2 (133.4, 143.0) | 137.6 (131.2, 144.0) | 138.1 (133.0, 143.2) | 0.219 |
| | K,(mmol/L) | | 4.4 (3.7, 5.1) | 4.4 (3.5, 5.3) | 4.4 (3.6, 5.2) | 0.485 |
| | Ca,(mmol/L) | | 8.4 (7.7, 9.1) | 8.3 (7.3, 9.3) | 8.4 (7.6, 9.2) | 0.136 |
| | Cl,(mmol/L) | | 102.5 (96.2, 108.8) | 100.1 (92.6, 107.6) | 102.1 (95.5, 108.7) | < 0.001 |
| | Glu,(mmol/L) | | 146.5 (78.4, 214.6) | 149.9 (87.3, 212.5) | 147.0 (79.7, 214.3) | 0.529 |
| | APTT,(s) | | 37.5 (14.7, 40.3) | 39.2 (18.0, 60.4) | 37.8 (15.3, 60.3) | 0.355 |

(*Continued*)

**Table 1.** (Continued)

| Variable | | | Survival(N = 1051) | Death(N = 183) | Total(N = 1234) | P-value |
|---|---|---|---|---|---|---|
| | INR | | 1.5 (0.8, 2.2) | 1.8 (0.3, 3.3) | 1.5 (0.7, 2.3) | 0.007 |
| | Cr,(mg/dL) | | 27.6 (5.2, 50.0) | 37.8 (12.3, 63.3) | 29.1 (5.9, 52.3) | < 0.001 |
| | BUN,(mg/dL) | | 1.4 (0.1, 2.7) | 1.9 (0.2, 3.6) | 1.5 (0.1, 2.9) | < 0.001 |
| | AG | | 14.5 (10.1, 18.9) | 16.4 (11.4, 21.4) | 14.8 (10.3, 19.3) | < 0.001 |
| **Comorbidities** | | | | | | |
| | Hypertension | | | | | 0.005 |
| | | No | 660 (62.8%) | 135 (73.8%) | 795 (64.4%) | |
| | | Yes | 391 (37.2%) | 48 (26.2%) | 439 (35.6%) | |
| | Myocardial infarction | | | | | 0.428 |
| | | No | 862 (82%) | 145 (79.2%) | 1007 (81.6%) | |
| | | Yes | 189 (18%) | 38 (20.8%). | 227 (18.4%) | |
| | Cancer | | | | | 0.867 |
| | | No | 841 (80%) | 148 (80.9%) | 989 (80.1%) | |
| | | Yes | 210 (20%) | 35 (19.1%). | 245 (19.9%) | |
| | Pneumonia | | | | | < 0.001 |
| | | No | 698 (66.4%) | 80 (43.7%) | 778 (63%) | |
| | | Yes | 353 (33.6%) | 103 (56.3%) | 456 (37%) | |
| | AF | | | | | 0.19 |
| | | No | 598 (56.9%) | 94 (51.4%) | 692 (56.1%) | |
| | | Yes | 453 (43.1%) | 89 (48.6%) | 542 (43.9%) | |
| | Renal Failure | | | | | < 0.001 |
| | | No | 618 (58.8%) | 49 (26.8%). | 667 (54.1%) | |
| | | Yes | 433 (41.2%) | 134 (73.2%) | 567 (45.9%) | |
| | CHD | | | | | < 0.001 |
| | | No | 585 (55.7%) | 136 (74.3%) | 721 (58.4%) | |
| | | Yes | 466 (44.3%) | 47 (25.7%) | 513 (41.6%) | |
| **Outcome** | | | | | | |
| | LOS in hospital, (days) | | 14.6 (3.0, 26.2) | 14.5 (1.5, 27.5) | 14.5 (2.4, 26.6) | 0.613 |

HR: Heart rate; RR: Respiratory rate; DBP: Diastolic blood pressure; SBP: Systolic blood pressure; MBP: Mean blood pressure; SPO2: Blood oxygen saturation; SOFA: Sequential Organ Failure Assessment; APSIII: Acute Physi ology Score III; SAPSII: Simplified Acute Phy siology Score II; GCS:Glasgow Coma Scale; CCI: Charlson Comorbidity Index; PLT:Platelet; HGB:Hemoglobin; AF:Atrial fibrillation; CHD: Coronary heart disease; Na: Sodium; K:Potassi um; Cl:Chlorine; Ca:Calcium; AG: Anion gap; Glu:Glucose; BUN:Blood urea nitrogen; Cr:Crea tinine; APTT: Activated partial thromboplastin time; INR: International normalized thewire.

NLR hospital mortality ($P < 0.001$) and there is a positive correlation between the trend of nonlinear correlation ($P < 0.05$), a nonlinear With growing levels of inflammatory markers, significantly increased their risk of death in hospitalized patients with; (S1 Fig in S2 File) in ICU mortality, only the SII ($P = 0.024$), SIRI ($P = 0.02$) and NLR ($P = 0.001$) showed similar results nonlinear ($P < 0.05$), SII, SIRI and NLR and trend of ICU mortality, there is a positive correlation between the nonlinear correlation.

ROC results showed that, In terms of in-hospital mortality and ICU mortality, SII (AUC = 0.637 and 0.623), SIRI (AUC = 0.659 and 0.636), PIV (AUC = 0.629 and 0.610), PLR (AUC = 0.617 and 0.592) and NLR (AUC = 0.676 and 0.671) on the prognosis of patients with COPD has certain prediction value, including NLR ability best (Fig 3A and 3B).

Next, we use KM curve analysis of 5 kinds of inflammatory markers and the prognosis of COPD inpatients survival situation. Research results show that (Fig 4), with the lowest quartile (Q1) as a reference, with the increase of level of inflammatory markers, the prognosis of

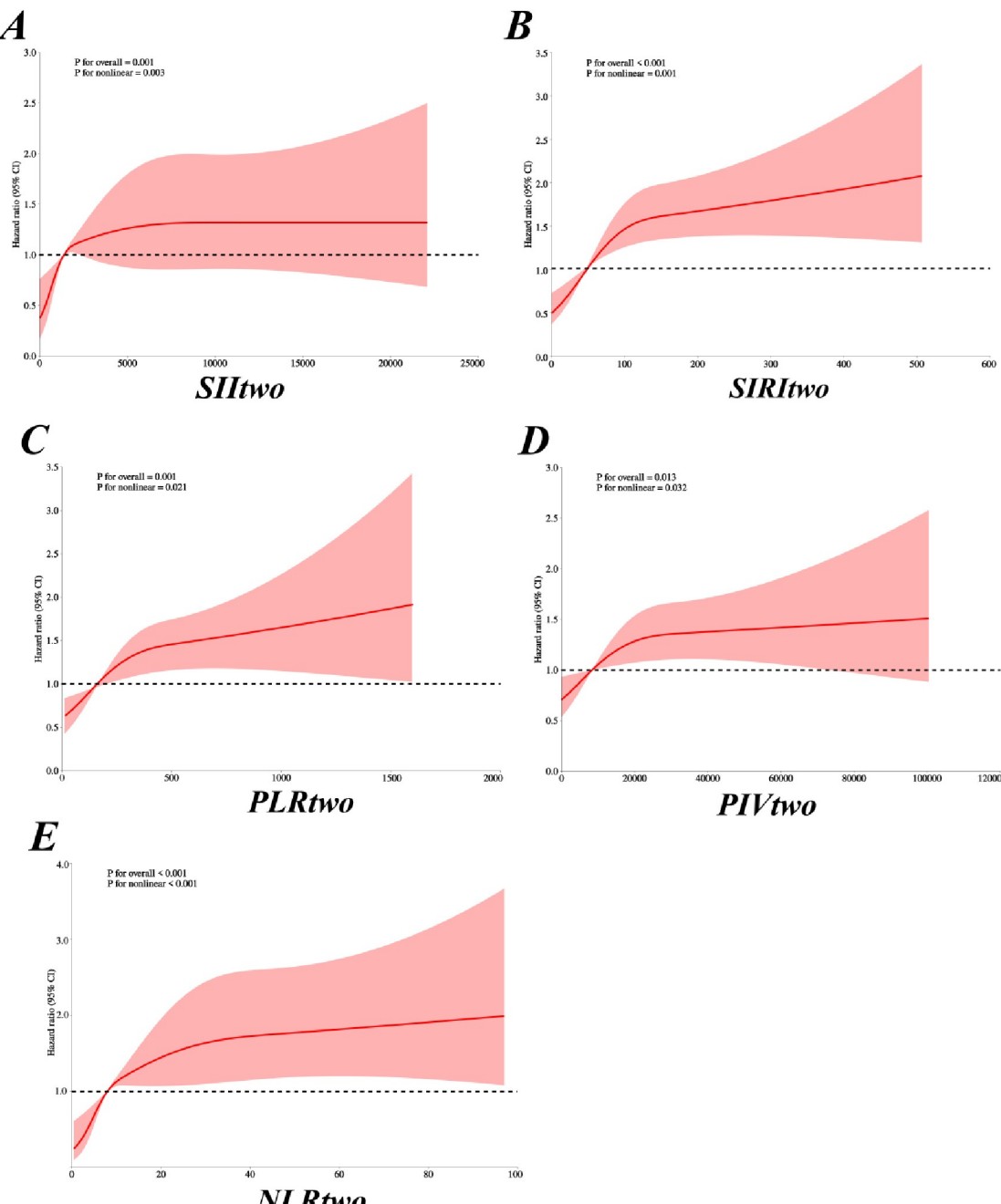

**Fig 2. Hospital mortality was restricted cubic spline figure.** (**A**) SII; (**B**) SIRI; (**C**) PLR; (**D**) PIV; (**E**) NLR.

patients with COPD survival may be falling, and SII, SIRI, PLR and NLR higher quarterback array (Q4) set of hospital survival rate was the lowest. Survival prognosis in the ICU (S2 Fig in S2 File), and the lowest quartile compared group (Q1), SII, SIRI and NLR rise in ICU patients with increased risk of death.

Cox proportional hazards regression models were used to evaluate the correlation between SII, SIRI, NLR, PLR and PIV and the risk of all-cause mortality in COPD patients, respectively. The results showed that, after adjusting for confounding factors, patients in the highest quartile

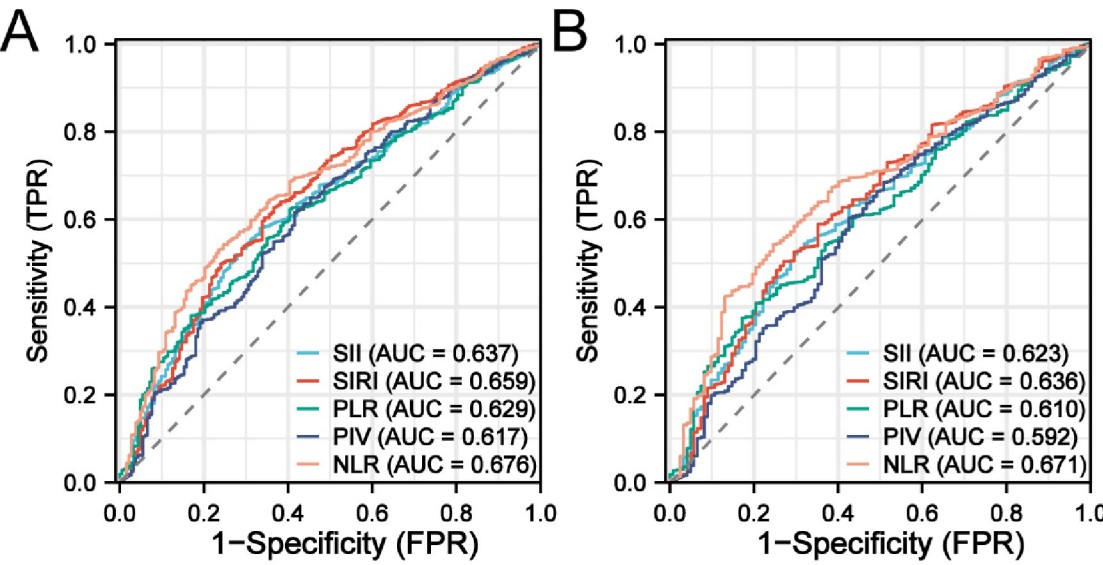

**Fig 3.** (**A**) the receiver-operating characteristic curve in-hospital mortality; (**B**) the ICU mortality.

(Q4 group) of SII (Table 2) exhibited a significantly increased likelihood of in-hospital death compared with those in the lowest quartile in four models constructed: Model I [HR (95%CI) 2.30 (1.42–3.72), $P < 0.001$], model II [HR (95%CI) 2.54 (1.54–4.18), $P < 0.001$], model III [HR (95%CI): 2.42 (1.47–4.00), $P = 0.001$) and the model IV [HR (95%CI): 2.24 (1.34–3.77), $P = 0.002$), and increase with the increase of the level of the risk with the SII. At the same time, in the model IV, SIRI [HR (95%CI) 1.77 (1.08–2.91), $P = 0.023$], the PLR [HR (95%CI) 2.76

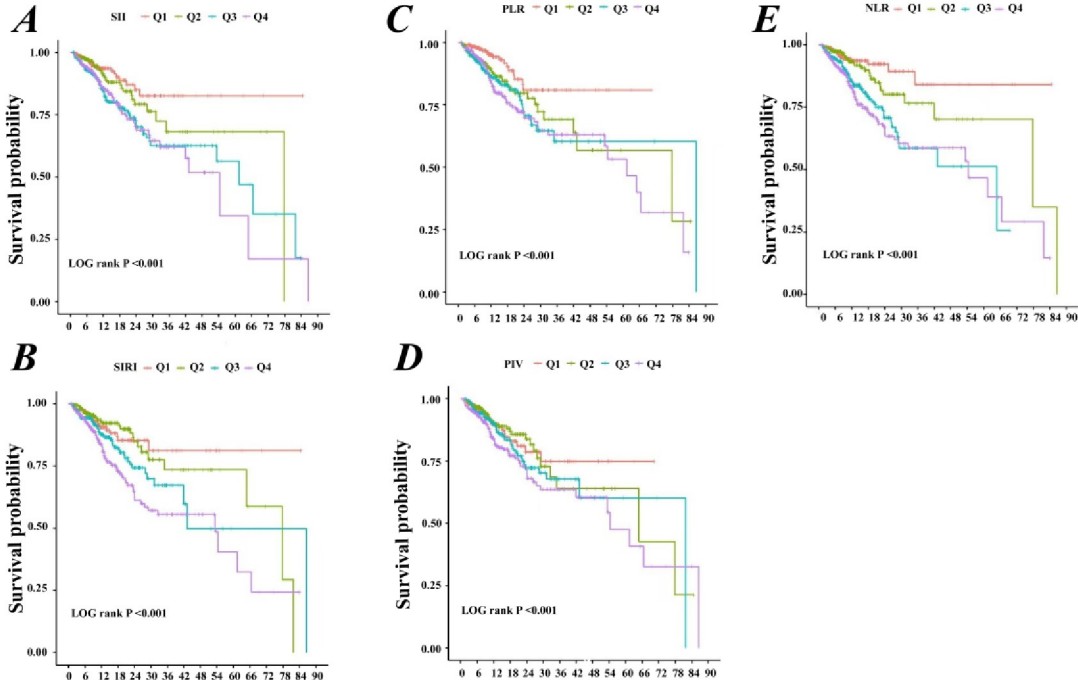

**Fig 4. Kaplan-Meier curves for hospitalized patients.** (**A**) SII; (**B**) SIRI; (**C**) PLR; (**D**) PIV; (**E**) NLR.

**Table 2. COX proportional hazards analysis of inflammatory markers (in-hospital mortality).**

| Variables | | Model I | | Model II | | Model III | | Model IV | |
|---|---|---|---|---|---|---|---|---|---|
| | | HR (95%CI) | P | HR (95%CI) | P | HR (95%CI) | P | HR (95%CI) | P |
| SII | | | | | | | | | |
| | Q1 | Ref. | | Ref. | | Ref. | | Ref. | |
| | Q2 | 1.39 (0.81–2.38) | 0.234 | 1.52 (0.88–2.65) | 0.135 | 1.46 (0.84–2.55) | 0.18 | 1.47 (0.83–2.59) | 0.187 |
| | Q3 | 2.34 (1.43–3.80) | < 0.001 | 1.9 (1.17–3.24) | 0.011 | 1.81 (1.08–3.04) | 0.024 | 1.93 (1.12–3.31) | 0.018 |
| | Q4 | 2.30 (1.42–3.72) | < 0.001 | 2.54 (1.54–4.18) | <0.001 | 2.42 (1.47–4.00) | 0.001 | 2.24 (1.34–3.77) | 0.002 |
| SIRI | | | | | | | | | |
| | Q1 | Ref. | P | Ref. | P | Ref. | P | Ref. | P |
| | Q2 | 1.02 (0.59–1.74) | 0.954 | 1.01 (0.58–1.74) | 0.978 | 0.98 (0.57–1.70) | 0.953 | 1.05 (0.59–1.84) | 0.878 |
| | Q3 | 1.65 (1.01–2.69) | 0.44 | 1.67 (1.02–2.74) | 0.043 | 1.55 (0.94–2.55) | 0.086 | 1.49 (0.89–2.49) | 0.132 |
| | Q4 | 2.46 (1.55–3.89) | < 0.001 | 1.94 (1.20–3.12) | 0.006 | 1.80 (1.11–2.90) | 0.016 | 1.77 (1.08–2.91) | 0.023 |
| PLR | | | | | | | | | |
| | Q1 | Ref. | P | Ref. | P | Ref. | P | Ref. | P |
| | Q2 | 2.03 (1.18–3.49) | 0.1 | 2.34 (1.35–4.05) | 0.002 | 2.13 (1.22–3.70) | 0.007 | 2.24 (1.27–3.97) | 0.006 |
| | Q3 | 2.35 (1.39–3.98) | 0.001 | 2.70 (1.56–4.65) | < 0.001 | 2.50 (1.44–4.34) | 0.001 | 2.78 (1.56–4.93) | < 0.001 |
| | Q4 | 2.61 (1.56–4.35) | < 0.001 | 2.62 (1.53–4.48) | < 0.001 | 2.45 (1.42–4.22) | 0.001 | 2.76 (1.56–4.89) | < 0.001 |
| PIV | | | | | | | | | |
| | Q1 | Ref. | P | Ref. | P | Ref. | P | Ref. | P |
| | Q2 | 0.99 (0.60–1.62) | 0.964 | 1.25 (0.75–2.07) | 0.394 | 1.27 (0.76–2.10) | 0.36 | 1.25 (0.75–2.09) | 0.388 |
| | Q3 | 1.27 (0.80–2.01) | 0.304 | 1.57 (0.98–2.52) | 0.059 | 1.51 (0.94–2.42) | 0.09 | 1.47 (0.90–2.39) | 0.123 |
| | Q4 | 1.61 (1.04–2.50) | 0.033 | 1.73 (1.09–2.76) | 0.02 | 1.63 (1.02–2.61) | 0.04 | 1.65 (1.01–2.69) | 0.044 |
| NLR | | | | | | | | | |
| | Q1 | Ref. | P | Ref. | P | Ref. | P | Ref. | P |
| | Q2 | 1.51 (0.83–2.75) | 0.175 | 1.48 (0.81–2.71) | 0.201 | 1.48 (0.81–2.72) | 0.2 | 1.48 (0.80–2.72) | 0.21 |
| | Q3 | 2.99 (1.74–5.12) | < 0.001 | 2.55 (1.48–4.40) | 0.001 | 2.51 (1.45–4.34) | 0.001 | 2.41 (1.37–4.24) | 0.002 |
| | Q4 | 3.51 (2.07–5.95) | < 0.001 | 2.35 (1.36–4.05) | 0.002 | 2.18 (1.26–3.78) | 0.006 | 2.25 (1.28–3.97) | 0.005 |

**Model I**: Unadjusted model; **Model II**: Age, gender, weight, race, HR, RR, body temperature, MBP, SBP, DBP, SpO$_2$, SOFA, SAPSII, APSIII, OASIS, GCS, CCI; **Model III**: On the basis of model II further adjustment of hypertension, myocardial infarction, malignancy, pneumonia, atrial fibrillation, renal failure and coronary heart disease; **Model IV**: WBC, RBC, central granulocyte count, lymphocyte count, monocyte count, PLT, HGB, Cl, K, Na, Ca, Glu, AG, Cr, BUN, APTT were adjusted on the basis of model III

(1.56–4.89), $P < 0.001$], PIV [HR (95%CI) 1.65 (1.01–2.69), $P = 0.004$) and NLR [HR (95%CI) 2.25 (1.28–3.97), $P = 0.005$] the highest quartile (Q4) arrays Cox proportional hazards model also showed similar results (Table 2).

**Table 3. COX proportional hazards analysis of SII (ICU mortality).**

| Variables | | Model I | | Model II | | Model III | | Model IV | |
|---|---|---|---|---|---|---|---|---|---|
| | | HR (95%CI) | P | HR (95%CI) | P | HR (95%CI) | P | HR (95%CI) | P |
| SII | | | | | | | | | |
| | Q1 | Ref. | P | Ref. | P | Ref. | P | Ref. | P |
| | Q2 | 1.26 (0.66–2.42) | 0.488 | 1.39 (0.71–2.71) | 0.335 | 1.28 (0.65–2.51) | 0.471 | 1.50 (0.74–3.02) | 0.259 |
| | Q3 | 1.83 (1.02–3.27) | 0.043 | 1.97 (1.08–3.60) | 0.028 | 1.81 (1.48–3.32) | 0.034 | 2.23 (1.13–4.39) | 0.041 |
| | Q4 | 2.11 (1.19–3.73) | 0.011 | 2.36 (1.29–4.34) | 0.006 | 2.13 (1.16–3.93) | 0.015 | 2.26 (1.20 4.29) | 0.012 |
| PLR | | | | | | | | | |
| | Q1 | Ref. | P | Ref. | P | Ref. | P | Ref. | P |
| | Q2 | 1.95 (1.02–3.73) | 0.044 | 2.27 (1.18–4.38) | 0.015 | 1.86 (0.95–3.64) | 0.069 | 2.20 (1.09–4.43) | 0.027 |
| | Q3 | 2.14 (1.13–4.05) | 0.019 | 2.32 (1.19–4.53) | 0.013 | 2.02 (1.03–3.95) | 0.041 | 2.69 (1.31–5.51) | 0.007 |
| | Q4 | 2.20 (1.18–4.10) | 0.014 | 2.44 (1.27–4.66) | 0.007 | 2.17 (1.13–4.19) | 0.021 | 2.72 (1.32–5.60) | 0.006 |

In terms of ICU mortality risk, the highest quartile of SII (Q4 group) appeared as an independent risk factor for mortality risk in COPD patients in all four models (Table 3). These models included unadjusted model 1 (HR (95%CI) 2.91 (2.51–3.39) $P < 0.001$), partially adjusted model 2 (HR (95%CI) 2.58 (2.22–3.00) $P < 0.001$), and partially adjusted model 3 (HR) (95%CI) 1.73 (1.48–2.02) $P < 0.001$], and fully adjusted model 4 [HR (95%CI) 1.39 (1.18–1.63) $P < 0.001$]. Similar results were observed in the Cox proportional-hazards model for the highest PLR quartile (group Q4) (Table 3).

## Subgroup analysis

In order to further analysis of complex inflammatory markers level influence to the prognosis of certain people, our main end different subgroups of patients with pneumonia composite inflammatory markers of risk stratification value assessment, including age, gender, weight, blood pressure, acute renal failure, malignant tumor, pneumonia, myocardial infarction, coronary heart disease and atrial fibrillation.

In-hospital mortality of subgroup analysis results show that the (Fig 5), inflammatory markers and age, sex, weight, blood pressure, malignant tumor, myocardial infarction, and no significant interaction between atrial fibrillation interaction ($P > 0.05$), and inflammatory markers in these subgroups are independent risk factors for in-hospital mortality in patients with COPD. SII (HR = 2.80, 95% CI = 1.59–4.93), PIV (HR = 2.19, 95% CI = 1.24–3.87) and PLR (HR = 2.16, 95% CI = 1.22–3.81) and no merger of renal failure patients with COPD is closely associated with an increased risk of death in hospital ($P < 0.05$); SII (HR = 2.35, 95% CI = 1.29–4.29), PIV (HR = 2.49, 95%CI = 1.35–4.58) and NLR (HR = 3.02, 95%CI = 1.67–5.48) were closely related to the increased risk of in-hospital death in COPD patients with coronary heart disease ($P < 0.05$); In COPD patients without pneumonia, SIRI (HR = 3.18, 95% CI = 2.05–4.93), PIV (HR = 2.78, 95%CI = 1.78–4.34) and PLR (HR = 1.94, 95%CI = 1.23–3.07) and hospital mortality significantly positive correlation between ($P < 0.05$); SIRI (HR = 1.56, 95%CI = 1.11–2.21) and merge kidney failure of hospitalized patients with COPD is closely associated with an increased risk of death, NLR (HR = 1.43, 95%CI = 1.01–2.02) and no merger of coronary heart disease (CHD) patients with COPD hospital is closely associated with an increased risk of death ($P < 0.05$).

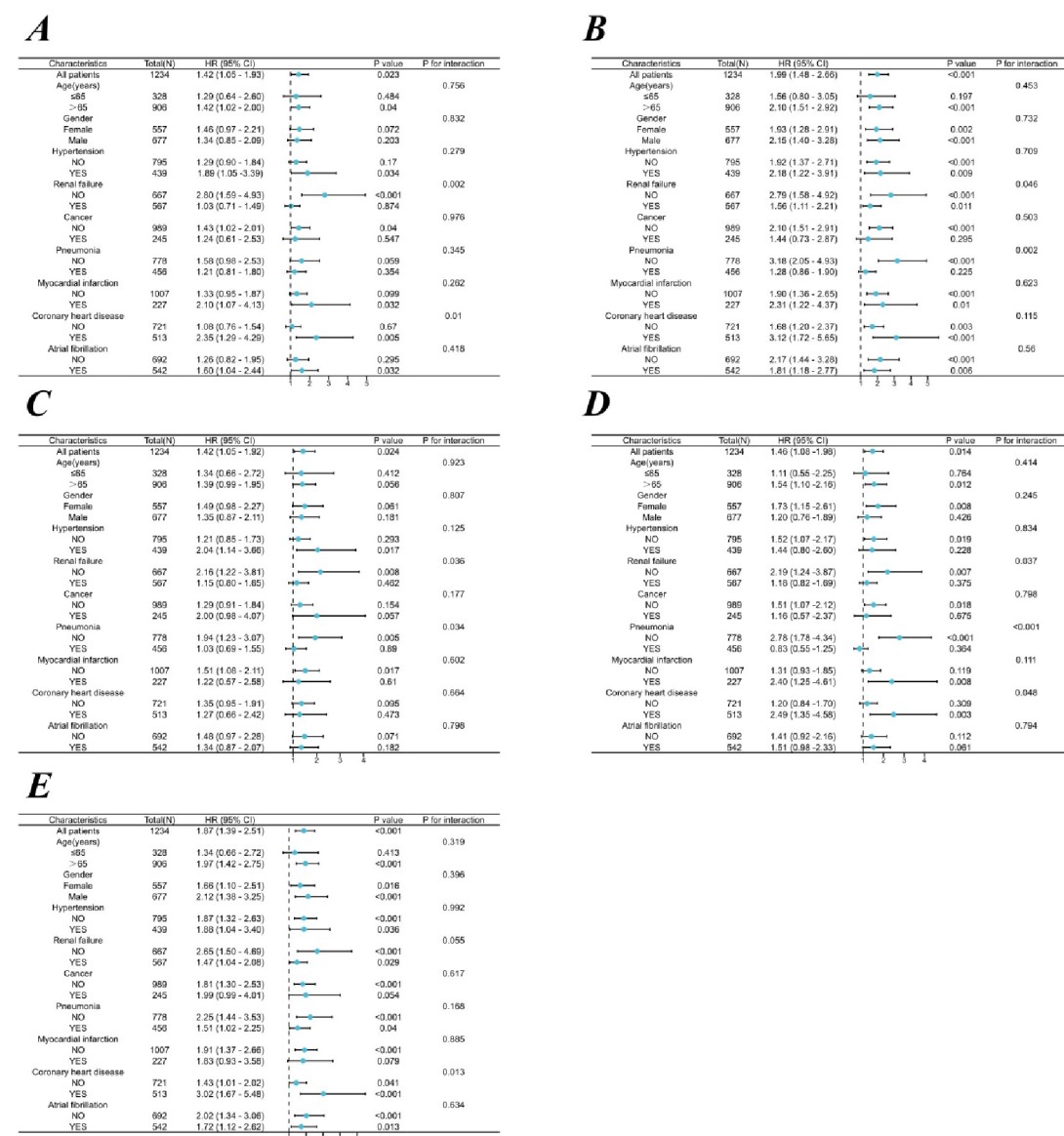

**Fig 5. Forest plot of subgroup analysis.** (**A**) SII; (**B**) SIRI; (**C**) PLR; (**D**) PIV; (**E**) NLR.

In terms of ICU mortality, subgroup results showed (S3 Fig in S2 File) that SII (HR = 2.97, 95%CI = 1.39–6.34), PIV (HR = 2.61, 95%CI = 1.55–4.38), and NLR (HR = 1.90, 95% CI = 1.25–2.88) increased the risk of death in COPD patients with concomitant renal failure, non concomitant pneumonia, and non concomitant myocardial infarction, respectively.

## Discussion

This study aimed to evaluate the role of SII, SIRI, PIV, NLR, and PLR5 composite inflammatory markers in predicting the risk of death in patients with COPD. In this retrospective cohort study, we confirm that the hospital mortality, ICU mortality in COPD patients with inflammatory markers were positive correlation tendency is nonlinear, mortality in patients with increased with elevated levels of inflammatory markers. The COX proportional hazards

regression analysis also showed that SII, SIRI, PIV, NLR, PLR can be regarded as independent risk factors for predicting the prognosis of patients with COPD.

COPD is characterized by persistent airway inflammation and immune dysfunction [6]. Relevant studies have shown that the levels of proinflammatory cytokines and oxidative stress in the airways of COPD patients are continuously reviewed, indicating that the airway inflammation state persists [6,8]. The disease development of COPD is associated with the change of immune system, including T cells (Th1) type 1 and type 2 T cells (Th2) between the change of the balance, as well as macrophages and dendritic cells and other immune cells number and function of change [4,9,10]. NLR represents the ratio of neutrophils and lymphocytes, clinically used to evaluate the body's inflammatory response and immune status, the higher as neutrophils, lymphocytes decreased with, may say the body is undergoing inflammation or immune imbalance, commonly used in the diagnosis and monitoring of infectious diseases. As a composite inflammatory marker, NLR plays an important role in monitoring acute exacerbations of COPD, and has been proved to be as valuable as CRP in assessing elevated inflammation in COPD, contributing to the clinical management [5] of COPD patients. Kurtipek proved that with the increase of NLR, such [11] as patients with COPD airway secretions increase, airway stenosis, shows that NLR is clinical markers of early identification of COPD worsens. Relevant studies have shown that NLR > 4.5 was found to be an independent risk factor for readmission of COPD patients due to acute exacerbation within one month, indicating that NLR can predict the short-term treatment effect [12] of COPD patients to a certain extent. Lee et [13] al. found that the NLR was associated with the severity of airflow limitation and that it could be used as a predictor of exacerbations and poor outcomes within 1 year. Two cohort studies showed that NLR had a prognostic effect [14,15] on mortality in patients with COPD.

Increase neutrophils is most of the patients with COPD airway inflammation and the features of lung inflammation means that patients with COPD in the stage of disease progression. The exacerbation of COPD, in addition to neutrophils, is always accompanied by an increase in the number of macrophages, lymphocytes, and dendritic cells. The composition of different inflammatory cells proportion changing according to the degree of disease progression. A retrospective study showed that patients with AECOPD and patients with stable COPD had higher levels of NLR [16] compared with healthy controls. NLR is a cost-effective indicators quickly, easily through routine laboratory tests [16] in clinical practice. Recently, studies have shown that NLR is associated [17–20] with the deterioration of COPD and severity. A 10-year prospective study included 386 cases to severe COPD patients, found that NLR is associated with long-term mortality in patients with COPD (HR = 1.090, 95%CI = 1.036, 1.148) [14]. Emami [21], points out that NLR acuity 6.9 patients mortality is higher than NLR < 6.9 patients. NLR is independently associated with in-hospital mortality in patients with AECOPD. It has long been recognized that the airways of patients with COPD and systemic inflammation associated [22,23] with disease progression, and mortality. Related studies have shown that lymphopenia is associated with high mortality in patients with systemic inflammatory response syndrome, and lymphopenia may perpetu [24,25] ate harmful inflammation. Lymphocyte count is negatively proportional to inflammation, while neutrophil count increases with the increase of systemic inflammatory response, which leads to a higher NLR and represents a poor prognosis of COPD patients to a certain extent. To sum up, NLR level at the same time provides the information of two different immune pathways, shows that the overall state of inflammation of the body, NLR is an independent predictor of mortality risk in patients with COPD, the levels are associated [26] with poor prognosis.

PLR represents the ratio of platelets to lymphocytes and reflects the balance between platelets and lymphocytes in an organism. It can be used to assess platelet activation induced by

inflammatory coagulation, severe coagulopathy, and systemic inflammatory responses. Most patients with COPD have hypofibrinolytic and hypercoagulable states, which are more likely to lead to thrombosis and increase the incidence of adverse prognosis events. Studies have shown that platelet activation can lead to structural changes in the pulmonary vascular system, and platelet activation [27,28] can be observed in patients with COPD and AECOPD. In the inflammatory response of COPD, pro-inflammatory cytokines stimulate megakaryocytes, leading to thrombocytosis. Platelets also bind to endothelial cells and promote leukocyte migration and adhesion to trigger and exacerbate inflammation [29]. Lymphopenia in inflammatory diseases is caused by the migration of lymphocytes to areas where inflammation occurs, leading to increased [23] ymphocyte apoptosis. Some studies have found that PLR levels are also significantly increased in patients with AECOPD compared with patients with stable COPD [11]. The increase in PLR not only reflects the imbalance between pro-inflammatory and anti-inflammatory responses, but also indicates the exacerbation of inflammatory responses.

The systemic immune inflammation index (SII) is a composite inflammatory biomarker that can comprehensively measure the systemic immune inflammatory response [30] of the human body. SII is a marker of chronic inflammation, which is manifested as increased neutrophil and platelet counts and decreased [31] lymphocyte count. Platelets participate in the inflammatory response, and activated platelets secrete a large number of inflammatory mediators, such as IL-1β and TNF-α, into the nearby microenvironment. These mediators can further promote the migration of white blood cells to the inflammatory site and accelerate the occurrence [32] of two major mechanisms of inflammatory reaction and oxidative stress in COPD. In addition, platelets also mediate the formation of pulmonary vascular microthrombi. Studies have shown that red blood cells in patients with COPD are closer to spherical than those in healthy people, which easily leads to the translocation of platelets to the vessel wall (marginalization) [33] and promotes the adhesion, aggregation and activation [34] of platelets on the vessel wall. The high aggregation of platelets may lead to arterial thrombosis, which further promotes the progression of COPD. Neutrophils are the most abundant white blood cells circulating in human blood and are considered to be the first line of innate immune defense. Their activation can release specific inflammatory mediators that cause irreversible airway damage, such as neutrophil elastase, matrix metalloproteinase-9 (MMP-9), cathepsin G, matrix metalloproteinase-48 (MMP-48), and myeloperoxidase (MPO), These substances can contribute to the pathophysiological mechanisms [35] of COPD. Neutrophil elastase stimulates mucin production and secretion, leading to mucus hypersecretion and airway obstruction [36], whereas MPO promotes oxidative tissue damage and initiates alterations in cellular homeostasis, and increases the response [37] of lung epithelial cells to proinflammatory stimuli. The inflammatory response of COPD may be the main cause of lymphopenia, and its mechanism is related to the apoptosis of lymphocytes when they resist the invasion of bacteria and virus antigens.

The calculation of PIV and SIRI involves neutrophils, monocytes, platelets and lymphocytes, which can be used as indicators [28,29] to evaluate different immune pathways and the inflammatory state of the body in patients. There is a close relationship between smoking and COPD, smoking is the most important risk factor for COPD, about 85–90% of cases of COPD is associated [38]with smoking. De et [39] al. proposed that cigarette smoke induces the release of damage-associated pattern molecules in a mouse model, which leads to innate immune responses and the release of interleukin-1β, monocyte chemoattractant protein-1, monocyte chemoattractant protein-2, and keratin chemokines, which further regulate the migration and infiltration of neutrophils and monocytes. Eventually lead to alveolar epithelial cell damage and death. Studies have shown that the level of monocyte chemoattractant protein-1 in

peripheral blood of COPD patients is higher than that of healthy people [40], suggesting that with the aggravation of COPD, the number of monocytes in peripheral blood of patients increases. At the same time, the number of lymphocytes in peripheral blood of COPD patients increased [41]. In the inflammatory response of COPD, neutrophils, platelets and monocytes show an upward trend, lymphocytes show a downward trend, PIV and SIRI levels continue to increase, and are closely related to the changes of COPD.

To the best of our knowledge, there is no evidence that the above five composite inflammatory markers affect the prognosis of COPD patients. We have demonstrated this association for the first time. Secondly, the sample size of this study is large and the data are from the real world, so the results are relatively reliable and have certain clinical guiding significance. Of course, there are some limitations of our study. First, this is an observational study, and there is no way for us to completely eliminate the possibility of bias or to definitively establish causality between variables and risk of death. Second, we studied only the first values after admission to the hospital or ICU and did not track dynamic changes in inflammatory markers that could have influenced the observed associations. Finally, despite the use of multivariable Cox proportional-hazards regression models, residual confounding and confounding by unquantified factors cannot be ruled out. Therefore, the results of the current study should be interpreted with caution, and additional multicenter prospective studies aimed at validating the relevance of composite inflammatory markers in predicting prognostic outcomes in patients with COPD are warranted.

## Conclusion

In conclusion, elevated SII, SIRI, PIV, NLR, and PLR are associated with an increased risk of short-term mortality in patients with COPD and may be potential novel markers for predicting mortality in patients with COPD. Five novel inflammatory markers are not only inexpensive and easy to obtain, but also help clinicians make timely judgments and interventions for the disease progression and survival prognosis of COPD patients. However, larger prospective studies with long-term follow-up are needed to confirm our conclusions.

## Supporting information

**S1 File.**
(XLSX)

**S2 File.**
(DOCX)

**S3 File.**
(XLSX)

## Author Contributions

**Writing – original draft:** Xingxing Liu, Yikun Guo.

**Writing – review & editing:** Wensheng Qi.

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
