## [Decision Letter · Decision Letter 0]

15 Sep 2024

PONE-D-24-36418Prognostic value of composite inflammatory markers in patients with chronic obstructive pulmonary disease: a retrospective cohort study based on the MIMIC-IV database

PLOS ONE

Dear Dr. Liu,

Thank you for submitting your manuscript to PLOS ONE. After careful consideration, we feel that it has merit but does not fully meet PLOS ONE’s publication criteria as it currently stands. Therefore, we invite you to submit a revised version of the manuscript that addresses the points raised during the review process.

Please provide a point to point response to the reviewers' comments. 

We look forward to receiving your revised manuscript.

Kind regards,

Chiara Lazzeri

Academic Editor

PLOS ONE

“Thank you for sponsoring the Science and Technology Innovation Project of Guang'anmen Hospital, Chinese Academy of Traditional Chinese Medicine.”

4. We note that your Data Availability Statement is currently as follows: [All relevant data are within the manuscript and its Supporting Information files]

5. Please upload a new copy of Figure 5 as the detail is not clear. Please follow the link for more information: https://blogs.plos.org/plos/2019/06/looking-good-tips-for-creating-your-plos-figures-graphics/" https://blogs.plos.org/plos/2019/06/looking-good-tips-for-creating-your-plos-figures-graphics/

Reviewers' comments:

Reviewer's Responses to Questions

**Comments to the Author**

1. Is the manuscript technically sound, and do the data support the conclusions?

Reviewer #1: Yes

Reviewer #2: Partly

2. Has the statistical analysis been performed appropriately and rigorously? 

Reviewer #1: I Don't Know

Reviewer #2: I Don't Know

3. Have the authors made all data underlying the findings in their manuscript fully available?

Reviewer #1: Yes

Reviewer #2: Yes

4. Is the manuscript presented in an intelligible fashion and written in standard English?

Reviewer #1: No

Reviewer #2: Yes

5. Review Comments to the Author

Reviewer #1: The authors showed through database analysis that five inflammation composite markers are useful for predicting survival in patients with COPD during admission to the ICU. The study was conducted on a relatively large number of patients. The results are reliable and clinically useful.

Reviewer #2: The study highlights the clinical significance of novel composite inflammatory markers like SII, SIRI, NLR, PLR, and PIV in predicting mortality in critically ill COPD patients. These markers provide valuable insights into the inflammatory response and immune dysregulation, essential for risk stratification and treatment management in acute settings. But I still have several questions:

1. How do the predictive values of SII, SIRI, PLR, and NLR compare to traditional markers of inflammation like CRP and procalcitonin in clinical practice for COPD management?

2. Could the findings of this study influence future clinical guidelines for the management of COPD, especially in the context of acute exacerbations and ICU admissions?

6. PLOS authors have the option to publish the peer review history of their article (what does this mean?). If published, this will include your full peer review and any attached files.

Reviewer #1: **Yes: **Kosho Yamanouchi

Reviewer #2: No

---

## [Author Response · Author response to Decision Letter 0]

4 Dec 2024

Dear Editor:

Thank you for your feedback and suggestions on my research manuscript. I have made modifications to the manuscript according to the PLOS ONE style template, removing any text related to fundraising and updating the relevant content in the cover letter. A new explanation regarding data availability has been listed in the attachment. Meanwhile, a clearer image of Figure 5 has also been uploaded in the manuscript for your reference. Thank you again for your review.

Reviewer 1:

Dear reviewer:

Thank you very much for taking the time to review this manuscript amidst your busy schedule. I have carefully read the suggestions you provided and I find them very useful. I highly appreciate your feedback and believe it has played a crucial role in improving this manuscript. I will further explore the clinical value of these inflammatory markers in future research, making their use more convincing.

Reviewer 2:

Dear reviewer:

I am very honored to have you review this manuscript. I have made detailed revisions to the relevant suggestions you have put forward, which I believe will provide great guidance for this manuscript. Thank you very much for taking the time to review it amidst your busy schedule.

Opinion 1: How do the predicted values of SII, SIRI, PLR, and NLR compare to traditional inflammatory markers such as CRP and procalcitonin in the clinical practice of treating chronic obstructive pulmonary disease?

Your first comment is of great clinical value, but unfortunately, there is no clinical information about procalcitonin (PCT) in patients in the MIMIC database. There are many missing information about C-reactive protein (CRP) in the relevant participants. Before analyzing in this study, it was considered that the missing values have a high risk of bias, so CRP data was excluded. Therefore, comparing the novel inflammatory markers with CRP and PCT through the MIMIC database cannot be achieved. However, when comparing the three and reviewing relevant literature, we found that SII, SIRI, PLR, and NLR not only reflect CRP and PCT levels to a certain extent [1], but also serve as auxiliary tools to help doctors diagnose and monitor the development of AE-COPD. On the other hand, NLR has the best predictive effect on the clinical prognosis and readmission risk of AECOPD patients, which is not inferior to CRP.

A study [2] found 112 patients diagnosed with AE-COPD, 92 stable COPD patients, and a control group consisting of 60 healthy individuals. Collect clinical features, CBC parameters, and serum CRP levels within two hours. Evaluate the association between NLR/PLR and CRP through Spearman correlation test. Evaluate the diagnostic accuracy of NLR and PLR in AE-COPD using receiver operating characteristic (ROC) and area under the curve (AUC). Perform binary logistic regression analysis on NLR and PLR indicators. Compared with stable COPD patients, AE-COPD patients have significantly higher levels of NLR and PLR. In addition, the study also revealed a significant correlation between CRP and NLR (rs=0.5319, P<0.001), PLR (rs=0.4424, P<0.001). The merging of NLR and PLR enhances diagnostic sensitivity by utilizing specific cutoff values. Binary logistic regression analysis indicates that elevated NLR is a risk factor for the progression of AE-COPD. In addition, Spearman correlation analysis showed a positive correlation between CRP and NLR/PLR. It can be seen that NLR and PLR can not only reflect CRP levels to a certain extent, but also serve as auxiliary tools to help doctors diagnose and monitor the development of AE-COPD. In addition, studies have shown that in all COPD patients, NLR values are positively correlated with serum CRP levels (r=0.641, p<0.001). For the NLR cut-off value of 3.34, the sensitivity of detecting COPD deterioration was 78.7%, and the specificity was 73.2% (AUC 0.863, p<0.001).The NLR, PLR, and lymphocyte to monocyte ratio in surviving AECOPD patients were lower than those in non surviving patients (all P<0.001). According to Pearson correlation test, there is a linear correlation between the new biomarker and the traditional biomarker (both P<0.05). As for a single biomarker, the AUC value of NLR is the highest, not inferior to C-reactive protein (Z-P=0.064), and better than new single inflammatory biomarkers such as erythrocyte sedimentation rate (Z-P=0.002) (all Z-P<0.05). The mortality rate of patients with NLR ≥ 4.43 is 2.308 times higher than that of patients with NLR<4.43. After dividing patients into high or low NLR groups, the summary results showed that patients with NLR ≥ 4.43 had a higher treatment failure rate, ICU admission rate, longer hospitalization time, one-year mortality rate after hospitalization, and overall mortality rate than patients with NLR<4.43 (all P<0.001). Compared with patients with lower NLR, patients with NLR ≥ 4.43 have a higher and earlier first-time readmission rate due to AECOPD. The predictive effect of NLR on the clinical prognosis and readmission risk of AECOPD patients is the best, not inferior to CRP, and the optimal critical value of NLR is 4.43 [4].

[1]Li L, Miao H, Chen X, etal. Research on the Correlation of Peripheral Blood Inflammatory Markers with PCT, CRP, and PCIS in Infants with Community-Acquired Pneumonia. Evid Based Complement Alternat Med. 2022 Oct 17;2022:9024969. doi: 10.1155/2022/9024969.

[2]Cai C, Zeng W, Wang H, Ren S. Neutrophil-to-Lymphocyte Ratio (NLR), Platelet-to-Lymphocyte Ratio (PLR) and Monocyte-to-Lymphocyte Ratio (MLR) as Biomarkers in Diagnosis Evaluation of Acute Exacerbation of Chronic Obstructive Pulmonary Disease: A Retrospective, Observational Study. Int J Chron Obstruct Pulmon Dis. 2024 Apr 15;19:933-943. doi: 10.2147/COPD.S452444. PMID: 38646605; PMCID: PMC11027921.

[3]İn E, KuluöztürkM,ÖnerÖ,DeveciF. The Importance of Neutrophil-to-Lymphocyte Ratio in Chronic Obstructive Pulmonary Disease. Turk Thorac J. 2016 Apr;17(2):41-46. doi: 10.5578/ttj.17.2.009. Epub 2016 Apr 1. PMID: 29404122; PMCID: PMC5792115.

Shao S, Zhang Z, Feng L, Liang L, Tong Z. Association of Blood Inflammatory Biomarkers with Clinical Outcomes in Patients with AECOPD: An 8-Year Retrospective Study in Beijing. Int J Chron Obstruct Pulmon Dis. 2023 Aug 17;18:1783-1802. doi: 10.2147/COPD.S416869. PMID: 37608836; PMCID: PMC10441637.

2. Opinion 2: Will the findings of this study affect future clinical guidelines for treating COPD, especially in cases of acute exacerbation and ICU hospitalization?

Thank you for the important questions raised by the reviewer. We believe that the study on the correlation between inflammatory factors SII, SIRI, PIV, NLR, and PLR with acute exacerbation of COPD and ICU hospitalization may indeed have an impact on future clinical guidelines.

As is well known, the clinical guidelines for COPD mainly include the following aspects: ① Diagnosis ② Treatment ③ Acute exacerbation management ④ Long term monitoring and follow-up. These guidelines aim to optimize the management of COPD patients, improve their quality of life, and reduce the risk of acute exacerbations. However, in clinical practice, the diagnosis of AE-COPD largely depends on the patient's medical history and clinical manifestations. However, the symptoms of AE-COPD are similar to those of various other diseases such as pneumonia and pulmonary embolism, posing challenges to the diagnostic process. In addition, in the early stage of AE-COPD, due to relatively mild symptoms, relying solely on clinical manifestations may not be accurate enough, which can easily lead to missed diagnosis or misdiagnosis. Therefore, at present, timely treatment and intervention through auxiliary examinations can quickly control the condition, alleviate symptoms, prevent further deterioration of the condition, and greatly improve the treatment effect of AE-COPD. Laboratory tests play a crucial role in identifying and diagnosing acute exacerbations of COPD. However, due to equipment limitations, the ability of grassroots and community hospital laboratories to test these methods is greatly restricted. Therefore, it is necessary to find easily accessible, affordable, and effective methods to assist in the diagnosis of AE-COPD. CRP and PCT are commonly used as indicators of inflammation in clinical practice, but due to various reasons, not all primary and community hospitals carry out CRP testing. Furthermore, compared to complete blood count (CBC), the cost of CRP testing is indeed higher. CBC is a fundamental clinical test that serves as a rapid laboratory aid and is also conducted in primary and community hospital laboratories.In the field of inflammation, lymphocytes and neutrophils are important indicators observed by CBC. CBC can reflect the systemic inflammatory response of different diseases. The presence of neutrophil lymphocyte ratio (NLR) is considered a hallmark of systemic inflammation. In addition, platelet lymphocyte ratio (PLR) has been evaluated as a predictor of various systemic inflammatory diseases. Compared with other common clinical inflammatory markers such as CRP and procalcitonin (PCT), CBC has the advantages of convenience and economy. Due to its fast and easy to access nature, it is the most common laboratory test, especially for primary and community hospitals. At present, the diagnosis of AE-COPD mainly relies on clinical symptoms and lacks specific serum biomarkers. However, if routine examinations of stable COPD patients reveal elevated levels of NLR, PLR, and MLR, it becomes crucial to remain vigilant about any changes in the patient's health status.

Our research confirms that the in-hospital mortality rate, ICU mortality rate, and inflammatory markers in COPD patients show a non-linear positive correlation trend, and the patient mortality rate increases with the level of inflammatory markers. SII, SIRI, PIV, NLR, and PLR can all serve as independent risk factors for predicting the prognosis of COPD patients. Our findings make up for the shortcomings of traditional detection methods and are convenient and easy to obtain. The use of inflammatory markers can improve the accuracy of predicting the exacerbation of acute COPD and patient prognosis, rather than relying solely on a single indicator. So the findings of this study are highly likely to affect future clinical guidelines for treating COPD, especially in cases of acute exacerbation and ICU hospitalization.

---

## [Editor Report · Decision Letter 1]

11 Dec 2024

Prognostic value of composite inflammatory markers in patients with chronic obstructive pulmonary disease: a retrospective cohort study based on the MIMIC-IV database

PONE-D-24-36418R1

Dear Dr. Liu,

We’re pleased to inform you that your manuscript has been judged scientifically suitable for publication and will be formally accepted for publication once it meets all outstanding technical requirements.

Kind regards,

Chiara Lazzeri

Academic Editor

PLOS ONE
---

## [Editor Report · Acceptance letter]

15 Jan 2025

PONE-D-24-36418R1 

PLOS ONE

Dear Dr. Liu, 

I'm pleased to inform you that your manuscript has been deemed suitable for publication in PLOS ONE. Congratulations! Your manuscript is now being handed over to our production team.

Kind regards, 

on behalf of

Dr. Chiara Lazzeri 

Academic Editor

PLOS ONE